

# Polar Firn Properties in Greenland and Antarctica and Related Effects on Microwave Brightness Temperatures

Haokui Xu[1], Brooke Medley[2], Leung Tsang[1], Joel T. Johnson[3], Kenneth C. Jezek[4], Macro Brogioni[5], and Lars Kaleschke[6]

[1]Radiation Laboratory, Department of Electrical Engineering and Computer Science, University of Michigan, Ann Arbor, MI 48105, USA
[2]Cryospheric Sciences Laboratory, NASA Goddard Space Flight Center, Greenbelt, MD, 20771, USA
[3]ElectoScience Laboratory, The Ohio State University, Columbus, OH, 43212, USA
[4]Byrd Polar and Climate Research Center, School of Earth Sciences, The Ohio State University, Columbus, OH 43210, USA
[5]Carrara Institute of Applied Physics, CNR, Florence, 50019, Italy
[6]Alfred-Wegener-Institut Helmholtz-Zentrum für Polar- und Meeresforschung, 27570 Bremerhaven, Germany

*Correspondence to*: Haokui Xu (xuhaoku@umich.edu)

**Abstract.** In studying the mass balance of polar ice sheets, the fluctuation of the firn density near the surface is a major uncertainty. In this paper, we explore these variations at locations in the Greenland Ice Sheet and at the Dome C location in Antarctica. Borehole in situ measurements, snow radar echoes, microwave brightness temperatures, and modelling results from the Community firn model (CFM) are used. It is shown that firn density profiles can be represented using 3 processes: "long" and "short" length scale density variations and "refrozen layers". Consistency with this description is observed in the dynamic range of airborne 0.5-2 GHz brightness temperatures and snow radar echo peaks in measurements performed in Greenland in 2017. Based on these insights, a new analytical partially coherent model is implemented to explain the microwave brightness temperatures using the three scale description of the firn. Short and long scale firn processes are modelled as a 3D continuous random medium with finite vertical and horizontal correlation lengths as opposed to past 1D random layered medium descriptions. Refrozen layers are described as deterministic sheets with planar interfaces, with the number of refrozen layer interfaces determined by radar observations. Firn density and correlation length parameters used in forward modelling to match measured 0.5-2 GHz brightness temperatures in Greenland show consistency with similar parameters in CFM predictions. Model predictions also are in good agreement with multi-angle 1.4 GHz vertically and horizontally polarized brightness temperature measured by the SMOS satellite at DOME C, Antarctica. This work shows that co-located active and passive microwave measurements can be used to infer polar firn properties that can be compared with predictions of the CFM. In particular, 0.5-2 GHz brightness temperature measurements are shown to be sensitive to long scale firn density fluctuations with density standard deviations in the range 0.01-0.06 g/cm^3 and vertical correlation lengths of 6-20 cm.



# 1 Introduction

The mass balance of polar ice sheets is a major topic in the study of climate change. The most recent assessment of the mass balance of the Greenland and Antarctic Ice sheets confirmed a loss of ice to the ocean at a rate of 320 Gt/year, equivalent to 1 mm sea level rise per year since 2003 [Smith et al 2020]. The quantification of uncertainty in ice-sheet volume change between NASA's first- and second-generation Ice, Cloud, and land Elevation Satellites (ICESat, ICESat-2) is a testament to the precision of these laser altimeters. For example, uncertainties for the grounded AIS and GrIS are currently ~5 and 3 $km^3$ yr-1, respectively, as compared to volume changes of -111 and -235 $km^3$ yr-1. At present, the largest source of uncertainty in altimetric measurements of mass balance stems from the volume-to-mass conversion within which firn processes dominate [Smith et al. 2020, Shepherd et al 2019].

When snow falls on the ice sheet, it slowly densifies into solid ice with increasing depth in a manner that is dependent on the pressure imparted by subsequent snowfall, the physical temperature, and any refreezing of infiltrated liquid water. The resulting transitional material is referred to as firn. Firn typically ranges in thickness from 10's to >100 meters over ice sheets (Ligtenberg et al., 2011; Kuipers Munneke et al 2015). The density of the firn column at a given location varies in time in response to short and long time-scale variations. Because the material density of the firn column is much less than that of solid ice, its thickness variations often manifest as a much larger portion of the total column thickness change than ice dynamic change. For example, firn density profiles in depth show fluctuations due to yearly snowfalls. The fluctuation amplitude becomes smaller and more rapid as depth increases because of densification effects.

Because of the large spatiotemporal variations in firn column properties, it can be extremely difficult to measure at the spatial scales required to support detailed modelling efforts. In situ measurements of the firn depth-density profile exist sporadically across both ice sheets in time and space [Montgomery et al., 2018]. While these observations provide a snapshot of firn properties, direct evidence of their evolution through time at sufficient resolutions applicable to altimetry studies (seasonally) remains a major challenge. Modelling efforts have attempted to fill in some of these knowledge gaps [Li and Zwally,2011; Kuipers Munneke et al, 2015], but their ability to realistically simulate firn processes remains incompletely understood in the absence of the extensive in-situ observations.

Active and passive microwave sensors can also inform us about the scattering and emission properties of the firn over large scales (Koenig et al 2007; Brucker et al., 2010; Champollion et al., 2013, Medley et al 2015); these properties are ultimately related to the physical properties of the firn. The strongest echoes in a radar echogram, for example, show the position of abrupt permittivity changes that usually correspond to the position of refrozen melt layers (Jezek and others, 1994; Zabel and others, 1995). Several studies have used active microwave remote sensing to track the internal stratigraphy (radar reflection horizons related to density contrasts) of the firn to infer spatiotemporal variations in snow accumulation rates [Medley et al., 2013; 2014; Koenig et al., 2016; Dattler et al., 2019]. Although radar echoes are able to position internal firn layers, using the radar data only to quantitatively study firn densification remains challenging.



Passive microwave brightness temperature measurements in the range 0.5-2 GHz can also reflect the effects of internal density fluctuations [Brogioni et al, 2015; Tan et al, 2019;]. Unlike radars which observe scattered powers only in the backscattered direction, radiometer brightness temperature observations are sensitive to scattering in all scattering directions within the firn
as shown by Kirchhoff's Law[Tsang 2001].

In this paper, we use co-located snow radar echoes (acquired in Greenland during the Operation Ice Bridge Campaign 2017, [CReSIS. 2021]) and 0.5-2 GHz brightness temperature data (the latter collected by the Ultra-Wide Band Software Defined Radiometer (UWBRAD) in 2017) to quantitatively evaluate firn density fluctuations in the Greenland ice sheet. In our previous works, we have used UWBRAD to sense the subsurface temperature profile, in which case the reflections caused by firn
density fluctuations are nuisance effects. In this paper, we instead treat these reflection effects as useful information to retrieve the frin density fluctuations. In addition to UWBRAD, we also use the multi-angle V and H pol SMOS brightness temperature data to evaluate the density fluctuations in Dome C, Antarctica. The firn density properties derived from microwave sensor data in Greenland are compared with simulated profiles from the Community Firn Model (CFM). The CFM was built as a resource to the glaciology community, and consists of a modular, open-source framework for Lagrangian modelling of several
firn and firn-air related processes (Stevens et al 2020). CFM simulation results can therefore represent firn density fluctuations at locations where the required input data fields are available. Firn density profiles predicted by the CFM have a vertical resolution that corresponds to the time sampling used in the model's simulations. Given the interest in short scale density functions in this paper, CFM predictions are generated in this work using a 5 day time step with seasonal initial density. To confirm the CFM predictions obtained, we compare simulation profiles with in situ measurements from ice cores and snow
pits, including the 1cm-resolved neutron probe data of Morris and Wingham (2011) collected near Summit station, Greenland in 2004. Coarser resolution but deeper in-situ profiles are also compared with CFM predictions for NEEM and NEGIS sites in Greenland. The CFM is also used to generate profiles for locations where snow radar and UWBRAD measurements are both available in Greenland, and at Dome C, Antarctica.

The snow radar dataset used has a bandwidth of 3.8 GHz to achieve a vertical resolution of 2 cm in the firn. Strong radar
echoes can be used to infer the locations and number of refrozen high-density layers within the firn. Refrozen layer locations inferred from snow radar measurements are shown to be consistent with those obtained from in-situ X-ray tomography data and CFM predictions.

Based on the radar, radiometer, CFM, and in-situ analyses reported, a firn density profile model is introduced that consists of three processes: "short" and "long" scale variations and "refrozen layers". The "short" and "long" scales in space are described
using differing shorter and longer vertical correlation lengths that correspond to "shorter" and "longer" time scale histories. The standard deviation of each density process and its correlation length are also decreased with depth to reflect firn compaction effects. Unlike the one-dimensional stochastic profiles used in previously brightness temperature modelling studies [Tan et al 2015, Tan et al 2020], a horizontal correlation length, $l_\rho$, is introduced for the short and long scale processes to





represent their variations in horizontal directions. This approach results in a continuous random medium description of the firn

as opposed to the past stochastic layered medium description. "Refrozen layer" effects (high or low-density density discontinuities) also were not included in [Tan et al 2015, Tan et al 2020], but are included in this paper.

A study of the effect of firn density variations on 0.5-2 GHz brightness temperatures is then performed using this description. An analytical partially coherent model of brightness temperatures is developed by applying the radiative transfer theory of microwave emission and scattering using a phase matrix corresponding to the continuous random medium description of the

firn density. The model then shows the effects of the long scale and refrozen layers to be significant, while those of the short scale process are negligible, and the impact of the long scale process is shown to depend on the microwave frequency. The number of freezing layers and their positions used in the model are determined from Greenland radar echo data. The results also show that freezing layers introduce a frequency dependence in 0.5- 2 GHz brightness temperatures that differs from that of the long scale process. Angular and polarization dependencies are investigated by comparing with SMOS data from Dome

C. The results in this case show that the horizontal correlation length $l_\rho$ impacts the coupling between V and H polarizations, and that a finite value of $l_\rho$ is required to obtain a good match to SMOS observations.

The model developed also suggests a means for combining active and passive microwave measurements to sense properties of firn density profiles in areas lacking in situ measurements. The method first estimates the number and location of freezing layers using radar echo measurements. The impact of these layers is then removed (based on the partially coherent model),

and properties of the long scale density fluctuations are estimated by matching model predictions to 0.5-2 GHz measured brightness temperatures. Results suggest that the long scale vertical correlation length can be estimated in this manner.

The next section provides an overview of firn density properties by comparing simulated CFM profiles with measured density profiles at the Summit, NEEM, and NEGIS sites in Greenland. Radar echo measurements are next compared with the corresponding CFM firn profile predictions and in-situ X-ray tomography measurements in Section 3 to investigate refrozen

layer effects. Section 4 then describes the new analytical partially coherent model for thermal emission, and simulated 0.5-2 GHz brightness temperatures are reported in Section 5 for locations in Greenland and at Dome C. Section 6 presents conclusions.

## 2. Firn density measurements at borehole sites and the associated CFM profiles

We use the Community Firn Model v1.1.6 (CFM; Stevens et al., 2020, 2021) to simulate the firn column density profile at

several locations across the ice sheet. CFM simulations are set up as detailed in Medley et al. (in Review) where the model is forced by a modified version of the MERRA-2 global atmospheric reanalysis (Gelaro et al., 2017) at 5-day temporal resolution. The only difference between the CFM simulations from Medley et al. (in Review) and those presented here is that a time-varying initial density $\rho_0$ of the firn column is introduced using the parameterization in Fausto et al. (2018): $\rho_0 = 362.1 + 2.78\,T_a$, where $T_a$ is the atmospheric temperature in °C at each time step. When comparing CFM-generated density profiles



with observations, we use the simulated profile that is most contemporaneous with the observations. For a detailed description
of the CFM set up, see Medley et al. (in Review). The vertical density profile of the firn can be characterized by $\rho(z) = \rho_m(z) + \rho_f(z)$, where $\rho_m(z)$ is a mean profile that gradually increases with depth and $\rho_f(z)$ is a fluctuating profile which
fluctuates around $\rho_m(z)$ and is characterized by standard deviation $\Delta\rho(z)$ and correlation length $l_z(z)$.

We selected 3 locations to compare in-situ measurements and CFM simulations. The first profile was collected at
T41(71.08N,37.92W) along the EGIG line by Morris [Morris and Wingham 2011] in 2004 using a neutron probe (Figure 1,
left). Data were collected up to 13 meters below the surface at a vertical resolution of 1 cm, and clearly show significant
fluctuations in density in the upper firn. The second profile and third profiles (Figure 2) are from a 2009 borehole measurement
at the NEEM site [Ian Baker. 2012. NEEM Firn Core 2009S2 Density and Permeability] with a vertical resolution of ~90 cm
and from a 2012 measurement at the NEGIS site having ~1 m increments. Firn density profiles from the CFM simulation are
also shown in Figure 1 and 2.

The high resolution profile at Site 1 enables an estimation of the mean profile and the standard deviation $std(\rho)$ and correlation
length $l_z$ of the fluctuating profile every meter in depth (Table 1). The coarser profiles at Sites 1 and 2 do not allow such
analysis, but information on the depth at which a "critical" density (i.e., 550 kg m$^{-3}$) is reached can be obtained.

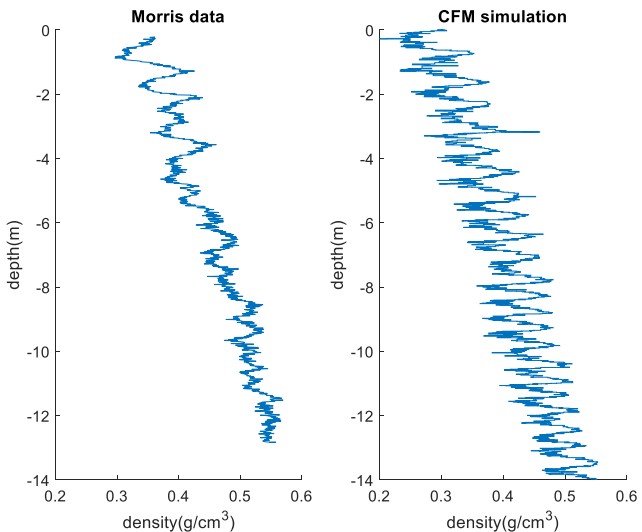

**Figure 1: (left) Morris and Wingham, 2011 density profile measured near Summit station, Greenland, Summer 2004
(right) corresponding CFM model simulation**






**Table 1:** Estimated density standard deviations (std(ρ)) and correlation lengths ($l_z$) estimated using 1 m of data beginning at the specified depth for Summit Station, Greenland from Neutron Probe dataset of Morris and Wingham, 2011 and the CFM

| Depth(z) | Neutron Probe data | | CFM | |
|---|---|---|---|---|
| | $std(\rho)(\frac{g}{cm^3})$ | $l_z$(cm) | $std(\rho)(\frac{g}{cm^3})$ | $l_z$(cm) |
| 0 | 0.028 | 16 | 0.03 | 9.6 |
| 2 | 0.019 | 17 | 0.027 | 12 |
| 4 | 0.01 | 12 | 0.029 | 11 |
| 6 | 0.012 | 17 | 0.028 | 9 |
| 8 | 0.011 | 12 | 0.028 | 7.5 |
| 10 | 0.0086 | 11 | 0.023 | 7 |

Both the in-situ and CFM profiles at Site 1 show small and fast variations superimposed on the larger but relatively slowly
varying mean profile. One-meter density standard deviations ($std(\rho)$) in Table 1 for the neutron probe and CFM are comparable, with most of the values around 0.03 g/cm^3. Vertical correlation lengths obtained both from the Morris' profile and the CFM simulation are <20 cm with mean values of 14.2 and 9.4cm, respectively.

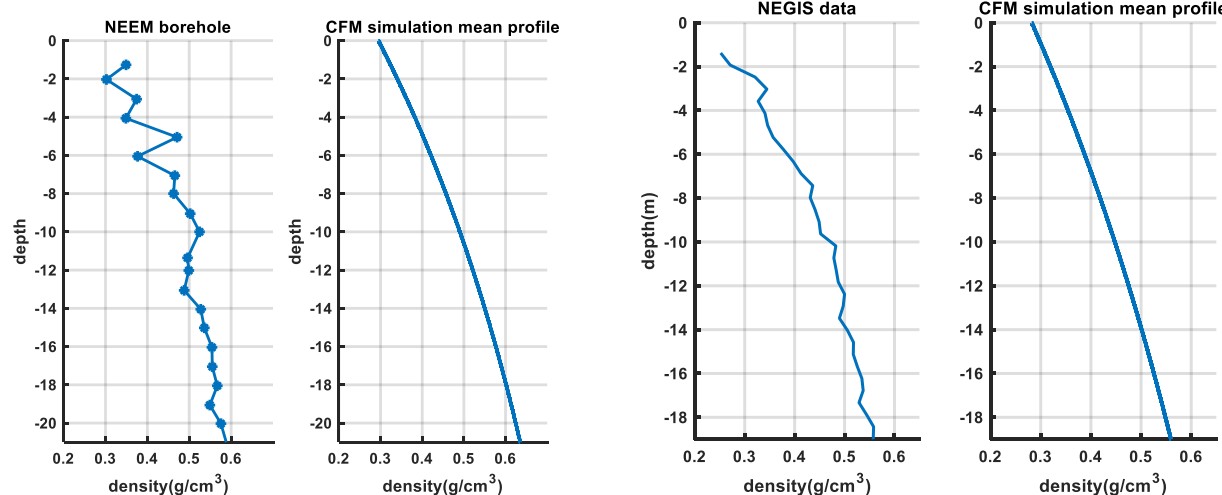

**Figure 2:** (left two plots) In-situ and CFM density profiles for the NEEM site (right two plots) In-situ and CFM density profiles
for the NEGIS site



**Table 2:** Estimated depth at which mean density profile reaches critical density value of 550 kg m$^{-3}$ for NEEM and NEGIS
In-situ Measurements and corresponding CFM simulations

| Site | Critical density depth: In-situ (m) | Critical density depth: CFM (m) |
|---|---|---|
| NEEM | 15.8 | 14 |
| NEGIS | 18.11 | 18.13 |

To estimate the critical density depth for profiles 2 and 3, we fit the data with an exponential function of depth that was then extrapolated to find the depth at which the exponential function reached 550 kg m$^{-3}$. The resulting depths from Profiles 2 and 3 and from the corresponding CFM simulations are shown in Table 2, and show reasonable agreement.

The results of this section suggest that the CFM, when run at a high time resolution, can produce firn density profiles that are in reasonable agreement with in-situ measurements.

## 3. Firn properties derived from microwave sensors over Greenland

UWBRAD measures ice sheet 0.5-2GHz brightness temperatures in a nadir viewing geometry. The cumulative effects of the temperature profile, density fluctuations, and the effects of refrozen layers are all included in the sensed brightness temperature. While previous studies have emphasized temperature profile sensing, UWBRAD could also be used to infer density fluctuations in the firn. The University of Kansas snow radar included in Operation IceBridge campaigns operates over the 2.5−6.3 GHz frequency range. Because the corresponding 3.8 GHz bandwidth enables a 2 cm vertical resolution of firn echoes, snow radar data can help to characterize near-surface properties of the firn. In particular, high dielectric contrast refrozen layers that extend over larger horizontal distances produce significant radar backscatter, enabling their characterization with radar measurements. The depths of refrozen layers within the firn can also be inferred based on the time delay of the associated radar echo. Radar measurements however are not optimal for sensing moderate density fluctuations within the firn because such fluctuations do not produce high backscattered power levels due to their low dielectric contrast.

To validate the potential utility of combined active and passive measurements of firn properties, locations one through four listed in Table 3 where snow radar and UWBRAD were nearly co-located over Greenland in 2017 were identified based on the 2017 flight paths shown in Figure 3. The first two locations are close to locations where X-Ray tomography of the firn was also performed.



| Overlapping points index (North to south) | Latitude | Longitude |
|---|---|---|
| 1 | 77.266N | 49.121W |
| 2 | 76.563N | 44.778W |
| 3 | 76.168N | 44.329W |
| 4 | 75.535N | 42.7948W |

**Table 3:** Latitude and Longitude for crossover points of 2017 UWBRAD and Snow Radar Measurements


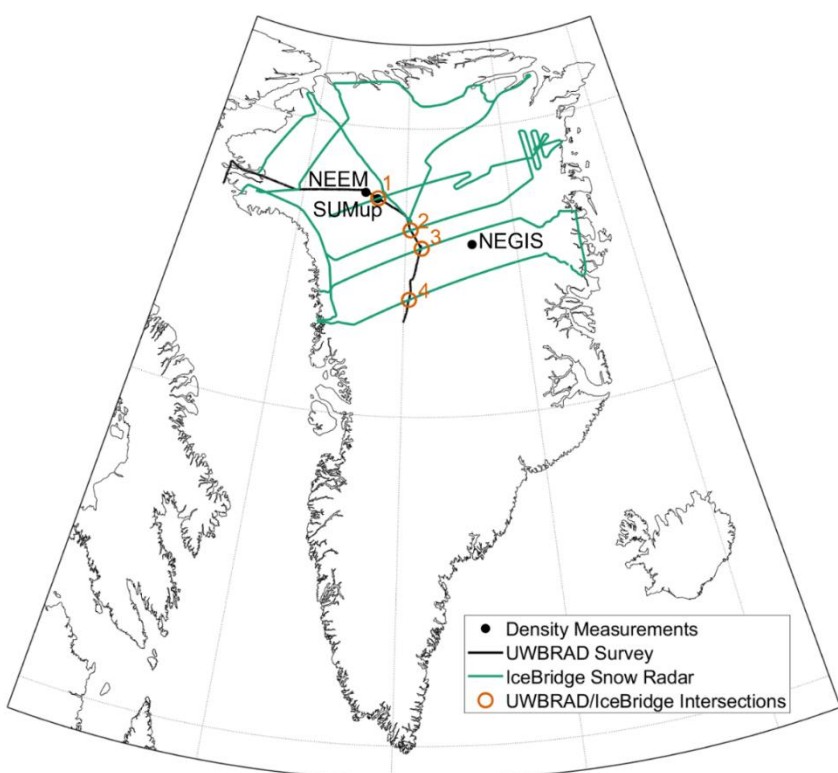

**Figure 3:** Flight path of Snow radar and UWBRAD measurements in 2017 with overlap points identified with circles.




Figure 4 plots three example radar echo profiles near Site 1 in Figure 3 along with an echogram showing multiple profiles versus position along the flight path. Individual echo profiles show multiple significant backscatter peaks in the upper firn, but the echogram demonstrates that returns can fluctuate significantly from location to location. An average of radar echoes over

a 1 km distance surrounding each site was therefore performed. CFM profile simulations also account only for large-scale climate properties in simulating firn profiles, so that a spatial average of radar measurements is also reasonable, and UWBRAD measurements also correspond to a footprint of 1 km diameter.

**Figure 4** (upper left) Snow radar echogram versus depth in the firn along the flight path. (Other plots) Three selected echo
profiles from the echogram. The echogram show bright edges near the surface which can be attributed to the refrozen layers with higher dielectric contrast.





Figure 5 and Figure 6 compare CFM simulated density profiles for cross-over sites 1-2 (Figure 5) and 3-4 (Figure 6) with the 1 km averaged radar echoes as a function of depth. All locations show secondary backscatter peaks at depth 2-2.5 m that correspond to peaks in the CFM density profiles. These peaks are due to a melt event that occurred in 2012 that affected much
of Greenland. The snow radar also observes backscatter peaks at shallower depths that do not correspond to similar density features in the simulated profiles, potentially due to inaccuracies in the climate forcing used for recent periods. Additional smaller backscatter peaks appear at 6-8 m depth that in many cases have matching CFM density peaks, but the lower level of the backscatter returns makes a direct comparison with CFM information more challenging.


**Figure 5 Averaged radar echos for cross over points one and two (upper left and right plots) and corresponding CFM simulated density profiles (lower left and right plots)**





**Figure 6 Averaged radar echos for cross over points three and two (upper left and right plots) and corresponding CFM simulated density profiles (lower left and right plots)**





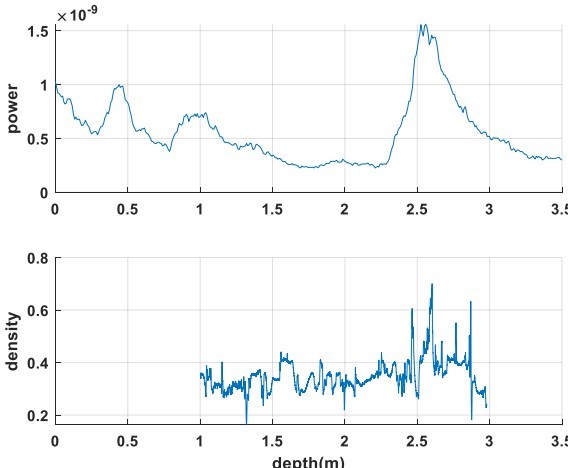

**Figure 7 Averaged snow radar echo (upper plot) compared to X-ray high resolution tomography density data (lower data) near cross-over site one**

2015 X-Ray tomography data providing a snapshot of the upper 2 meters of the firn for a location near cross-over point 1 is shown in Figure 7. The X-ray profile was shifted 1 meter in depth to compensate for snow fall between the 2015 tomography and 2017 radar measurements. The strong radar echo near 2.5 depth again collocates well with X-ray density features near this depth related to the 2012 melt event. It is noted that the tomographic profile corresponds only to a single location rather than the 1 km average used in the radar echo, so that the detailed features in the tomography profile are not observed in the averaged radar measurement, although similar detailed features can be identified in some individual radar echo profiles. Table 4 presents a summary of the numbers of peaks detected in the averaged snow radar echogram at each of the four cross-over sites.

**Table 4:** Peaks in snow radar echoes

|                  | Point1 | Point 2 | Point 3 | Point 4 |
|------------------|--------|---------|---------|---------|
| Number of peaks  | 3      | 2       | 2       | 2       |

The results of this section suggest that strong echoes observed by the snow radar are due to refrozen layers in the firn. CFM simulated density peaks are shown to correspond reasonably to high backscatter echoes, with clear evidence of refrozen layers caused by the summer 2012 melt event. X-Ray tomography data also shows the impact of the 2012 melt event and correlates reasonably with radar measurements.

## 4. Analytical Partially coherent model

An illustration of ice sheet thermal emission problem is shown in Figure 8:. A firn layer of thickness $d_1$ (region 1) exists near the ice sheet surface. The density of the firn layer is modelled as $\rho(\vec{r}) = \rho_m(z) + \rho_f(\vec{r})$ with $\rho_m(z)$ the mean density



profile. The fluctuating profile is described as $\rho_f(\vec{r}) = \rho_{fs}(\vec{r}) + \rho_{fl}(\vec{r})$. Notice that the fluctuating profile varies in three

dimensions and has two scales, short ($\rho_{fs}$) and long ($\rho_{fl}$). The real and imaginary parts of the microwave permittivity of the

firn are related to the firn density using the models in [Matzler, Tiuri]. The correlation function for each scale of the fluctuating

density is described by

$$(\Delta\varepsilon_{rf})^2 C(|\vec{r} - \vec{r}'|) = (\Delta\varepsilon_{rf}(z))^2 \exp(-\frac{|z - z'|}{l_z(z)}) \exp(-\frac{(x - x')^2 + (y - y')^2}{l_\rho^2})$$

in which $\Delta\epsilon_{rf}(z)$ is the standard deviation, $l_z(z)$ is the permittivity vertical correlation length, and $l_\rho$ is the horizontal

correlation length. The correlation function is described as having a Gaussian form laterally and an exponential form vertically

based on the model used in [Tsang 2001]. The exponential form for the vertical correlation function is adopted based on

analyses showing similar properties for the firn density itself. Both $\Delta\epsilon_{rf}$ and $l_z$ are modelled as functions of depth due to the

compaction of the firn, while $l_\rho$ is modelled as independent of depth.

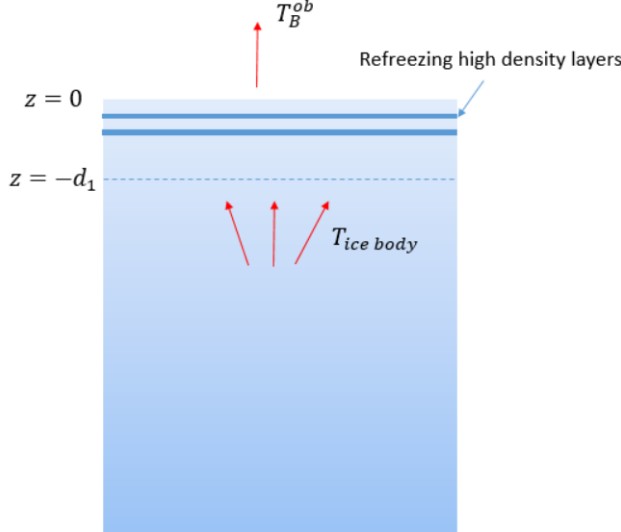

**Figure 8:** Thermal emission from an ice sheet

Below region 1, the main ice body can be multiple km thick with a temperature profile that varies in depth. Thermal emission

from the main ice body is calculated using the existing partially coherent model [Tan et al 2020] and the temperature profile

obtained in [Yardim et al 2022]. The temperature in region 1 is modelled as a constant value $T_0$ . Although the top 10-20 m of

firn experiences seasonal temperature changes, these variations have little effect on brightness temperatures at frequencies less

than 2 GHz due to the limited emission directly from the firn layer.



Applying the radiative transfer theory of microwave emission and scattering, the upward and downward propagating specific intensities $\vec{I}_u$ and $\vec{I}_d$ in region 1 satisfy

$$\cos\theta\frac{d}{dz}\vec{I}_u(\theta,z) = -\kappa_a(z)\vec{I}_u(\theta,z) - \overline{\overline{\kappa}}_s(\theta,z)\vec{I}_u(\theta,z) + \kappa_a(z)C\vec{T}_0 + \int_0^{2\pi}\sin\theta'd\theta'\overline{\overline{F}}(\theta,\theta',z)\vec{I}_u(\theta',z) + \int_0^{2\pi}\sin\theta'd\theta'\overline{\overline{B}}(\theta,\theta',z)\vec{I}_d(\theta',z)$$

$$-\cos\theta\frac{d}{dz}\vec{I}_d(\theta,z) = -\kappa_a(z)\vec{I}_d(\theta,z) - \overline{\overline{\kappa}}_s(\theta,z)\vec{I}_d(\theta,z) + \kappa_a(z)C\vec{T}_0 + \int_0^{2\pi}\sin\theta'd\theta'\overline{\overline{B}}(\theta,\theta',z)\vec{I}_u(\theta',z) + \int_0^{2\pi}\sin\theta'd\theta'\overline{\overline{F}}(\theta,\theta',z)\vec{I}_d(\theta',z)$$

with the boundary conditions:

$$\vec{I}_d(\theta,z=0) = \overline{\overline{r}}_{10}(\theta)\vec{I}_u(\theta,z=0)$$

$$\vec{I}_u(\theta,z=-d_1) = C T_2$$

In the above equations, $\vec{I}_u$ and $\vec{I}_d$ are 2 x1 vectors in which the upper row is for vertical polarization and the lower row for horizontal polarization. $\kappa_a(z)$ is the absorption coefficient determined by the mean density profile while $\overline{\overline{\kappa}}_s(\theta,z)$ is the scattering coefficient due to the randomly fluctuating portion of the density profile. The phase matrices $\overline{\overline{F}}$ and $\overline{\overline{B}}$ couple specific intensities from other directions $\theta'$ into the direction of interest $\theta$ in the forward or backward propagating

hemispheres. The boundary conditions specify that the firn-to-air interface at $z=0$ is reflective with reflection coefficient $\overline{\overline{r}}_{10}(\theta)$ and that the compacted firn to ice interface at $z=-d_1$ with $d_1=100m$ is not reflective. An iterative approach is then used to solve the equations. Since the permittivity variation is small, the first order solution together with the zeroth order solution provides sufficient accuracy. A detailed solution of the equations can be found in Appendix A. The method is partially coherent because the phase matrices are obtained using a coherent formulation of the continuous medium scattering problem.

High-density refrozen layers are included by incorporating their additional reflections as

$$T_b^{ob} = (1-r_{10})\frac{\lambda^2}{K}\frac{1}{\varepsilon'_{rm1}}(I_u^{(0)}(\theta=0,z=0)+I_u^{(1)}(\theta=0,z=0))\prod_n(1-r_n^{refrez})$$

where $\prod_n(1-r_n^{refrez})$ accounts for the transmission from each layer. This multiplicative approach is reasonable because the microwave wavelength between 0.5~2GHz is larger than the typical layer thickness.

The resulting model captures coupling between scattering in different directions and polarizations through the phase matrices $\overline{\overline{F}}$ and $\overline{\overline{B}}$. The previous "random layer" 1D formulation of Tan et al 2020 captures neither of these effects.




## 4.1. Studies of the impact of each density component on 0.5-2 GHz brightness temperatures

The model was first applied to simulate the impact of "long scale" density fluctuations on 0.5-2 GHz brightness temperatures.
Figure 9: shows example reflections resulting from long scale firn density variations at the snow-air interface, using the
parameter in Table 5. The maximum and minimum of the reflection within the bandwidth is 0.126 and 0.118, a difference of
0.008. The reflectivity is significant but remains approximately constant in frequency in this case. A similar study for short
scale variations using the parameters in Table 5 is presented in Figure 10: . The small reflectivity values obtained suggest that
the contribution of short scales is negligible compared to the long scale.

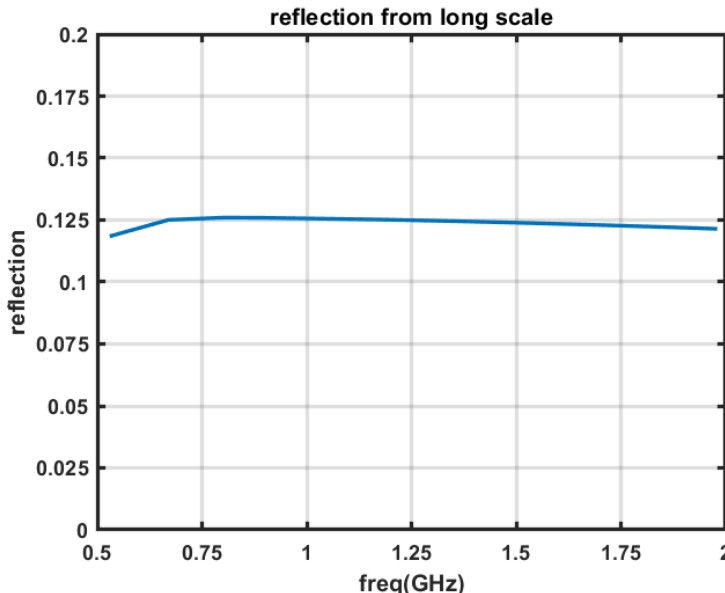

**Figure 9:** Reflections from the long scale and snow-air interface; the results are almost constant in frequency





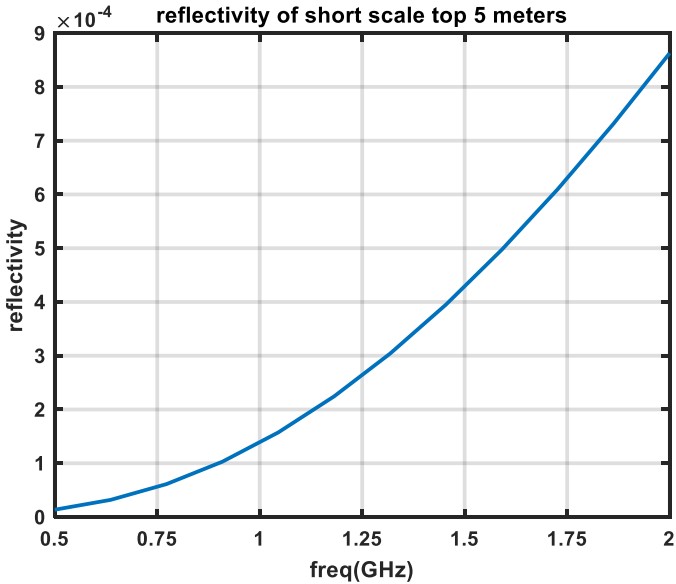

**Figure 10:** Reflectivity from short scale fluctuations in the top five meters

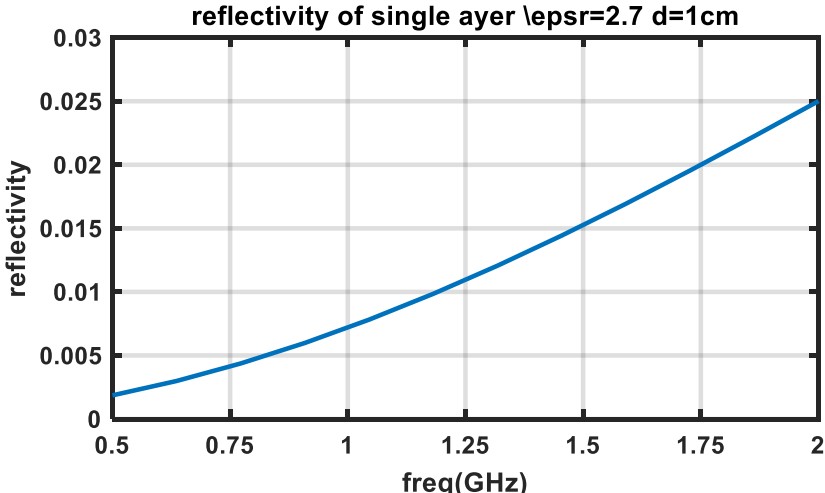

**Figure 11:** Reflectivity of a single layer with permittivity 2.7 and 1cm thickness in a mean permittivity of $\epsilon_r = 1.63$(0.35g/cm^3 in density)

The reflectivity resulting from a single refrozen layer of relative permittivity 2.7 and 1 cm thickness in a background of relative permittivity 1.63 is shown in Figure 11. These results show a significant variation with frequency, ranging from 0.002 to 0.025 from 0.5 to 2GHz, suggesting that refrozen layers can be important contributors to the frequency variation in 0.5-2 GHz brightness temperatures. Table 6 provides a further summary of insights obtained from Figures 9-11 and other similar simulations.



**Table 5:** Properties of the short scale, long scale variations in density and high-density layers

| Scales Deterministic or random 3D or layered measurements | $corl_\rho$ (not modelled, a hypotheses), $l\_\rho/l\_z$ | $\Delta\rho(g/cm^3)$ $corl_z$ or thickness(cm) | Extent in depth | Number of reflections | Reflections Magnitude (i) each (ii) Total | Included in Community Firn Model (CFM)? |
|---|---|---|---|---|---|---|
| Short scale (random 3D) borehole | $2\ cm$ $\dfrac{l_\rho}{l_z}=1$ | $0.01\ g/cm^3$ $2\ cm$ | 5m | 250 | 2.5e-5 0.00625=250x2.5e-5 | Not modelled because of small variance and small correlation length |
| Long scale (random 3D) borehole | 23 cm $\dfrac{l_\rho}{l_z}=2.3$ | $0.05\ g/cm^3$ $10\ cm$ | 30m | 300 | 5e-4 0.15=300x5e-4 | yes |
| Frozen layers (deterministic, 1D ) From radar echo in time domain | $l_\rho > 10\lambda$ | $0.3g/cm^3$ 1cm | 2m | 3 | [0.002~0.025] 0.006~0.075 | yes |

**5.0 Comparisons of Modelled and Measured Brightness Temperatures**

UWBRAD dataset: The analytical partially coherent model was first used to simulate UWBRAD acquired brightness temperatures.  Figure 12 plots UWBRAD 0.5, 1.1, and 1.8 GHz brightness temperatures at the four cross-over locations. The frequency variations of brightness temperatures at point one (right most point in Figure 12) are larger than the other three points, which can be related to the number of radar echo peaks at these locations (Table 4).



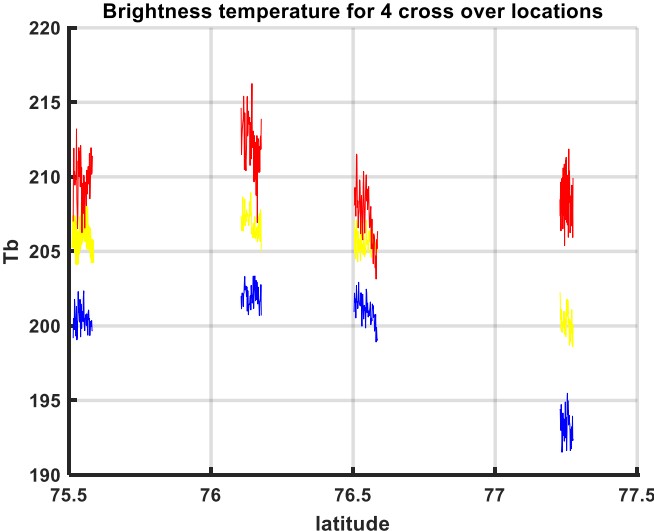

**Figure 12:** Brightness temperature for 3 channels of 0.5GHz (red), 1.1GHz (yellow) 1.8GHz (blue). Cross-over locations 1 to 4 are from right to left in the figure.

To simulate UWBRAD brightness temperatures, temperature profiles from NGRIP to NEEM retrieved in [Yardim et al 2022] (see Appendix B) are used with the partially coherent model [Tan et al 2020] to provide the upward going brightness temperature at depth $z = -d$. A mean profile of $\rho_m(z) = 0.917 - 0.5748\exp(0.0263z)$ is also used for all four cross-over locations based on analysis of CFM outputs for the 4 locations which were found to have similar mean density behaviours. An iterative process was used to refine model parameters in order to obtain a reasonable match to the UWBRAD measurements, as shown in Figure 13. It is noted that UWBRAD variations at these sites can be approximately 3K, so that the agreement achieved is comparable to the measurement accuracy.

Table 6 summarizes the parameters obtained; note that $std(\rho)$ and $l_z$ were decreased in depth through a multiplication with the functions $\exp(z/33)$ and $\exp(z/55)$, respectively where z is the depth in meters, for the large scale and by $\exp(z/5)$ for the short scale. The horizontal correlation length for the long scale variations was obtained as $l_\rho = 23cm$, which appears consistent with reports from in-situ investigations and is similar to the properties of firn surface horizontal variations. The permittivity and thickness of the high-density layers used in simulating the brightness temperature are also listed in Table 6; the number of high-density layers at each site was selected based on the radar analysis in Table 4.

Long scale density fluctuation parameters inferred from the CFM are listed in Table 7 for comparison. For site one, the microwave estimated $std(\rho)$ is 0.058g/cm^3 with $l_z = 11.5cm$, while the corresponding values for sites two through four are 0.053g/cm^3 with correlation length ~ 9cm. The CFM results show $std(\rho)$ values of 0.036g/cm^3 with $l_z = 10.5cm$ for Pt1 and 0.033g/cm^3 with vertical correlation length close to 7cm for the other sites. While the $std(\rho)$ used in the forward model is about 0.02g/cm^3 higher than the CFM, $std(\rho)$ values in both cases agree in the higher $std(\rho)$ and $l_z$ values at site 1. While differences in the microwave-derived and CFM derived density fluctuation values are significant, the





relative agreement achieved suggests that 0.5-2 GHz brightness temperatures can provide information on firn density fluctuations if refrozen layer effects are accounted for using radar-derived information.

**Figure 13:** Brightness temperature over the 4 overlapping positions. The simulated results are plotted together with the UWBRAD data






**Table 6:** Parameters used in forward modelling brightness temperature, the decrease of $std(\rho)$ and $l_z(z)$ follows $exp(z/33)$

|  |  | Pt1 | Pt2 | Pt3 | Pt4 |
|---|---|---|---|---|---|
| Long scale | $std(\rho)(z = 0)$(g/cm^3) | 0.058 | 0.053 | 0.053 | 0.054 |
|  | $l_z(z = 0)(cm)$ | 11.5 | 9.1 | 9.3 | 9.2 |
| Refrozen layers | Permittivity | 2.6;2.7;2.7 | 2.6;2.7 | 2.6;2.7; | 2.6;2.6 |
|  | Thickness (cm) | 0.9,1.1,1.1 | 0.95,0.9 | 0.95,0.9 | 0.85,0.9 |


**Table 7:** Near surface long scale properties from CFM simulation and Accumulation rate

|  | Pt1 | Pt2 | Pt3 | Pt4 |
|---|---|---|---|---|
| $std(\rho)$(g/cm^3) | 0.036 | 0.033 | 0.0325 | 0.0325 |
| $l_z$(cm) | 10.5 | 6.2 | 7.14 | 7.2 |
| Accumulation (m i.e. yr[-1]) | 0.293 | 0.148 | 0.148 | 0.193 |

**SMOS observations over Dome C (75∘ S, 123∘ E), Antarctica**: Forward model predictions were also compared with brightness temperatures at Dome C measured by the SMOS satellite. SMOS operates at 1.4 GHz and has both V and H channels. The synthetic aperture technique in SMOS make multi-angle observations possible as provided in the SMOS L1C

data product. Firn properties at DOME C are very different from those in Greenland. The accumulation rate of Dome C is 0.1m/year [Brogioni et al 2015] in contrast to the higher accumulation rates in Greenland shown in **Table 7**. A shorter correlation length therefore should be expected as compared to Greenland, and temporal effects on the firn will less significant compared to Greenland. Refrozen layers at this site are also neglected as significant melt events are not expected. A comparison of the firn properties used in Greenland and at Dome C is provided in Table 8 to summarize these discussions.


**Table 8:** Firn properties in Greenland and at Dome C

| Scales | Greenland | Dome C |
|---|---|---|
| Short scale(z=0) | $0.01g/cm^3, l_z = 2cm, l_\rho = 2cm$ | No short scale |
| Long scale(z=0) | $0.05g/cm^3$ , $l_z = 10cm$, $l_\rho = 23cm, \frac{l_\rho}{l_z} = 2.3$ | $0.04g/cm^3$ , $l_z = 5.5cm$ , $l_\rho = 23cm$ |





| Frozen layers | $0.3 g/cm^3$ | No Frozen layers |
| --- | --- | --- |
| | 1cm thick, $l_\rho > 10\lambda$ | |
| | 3 layers, | |

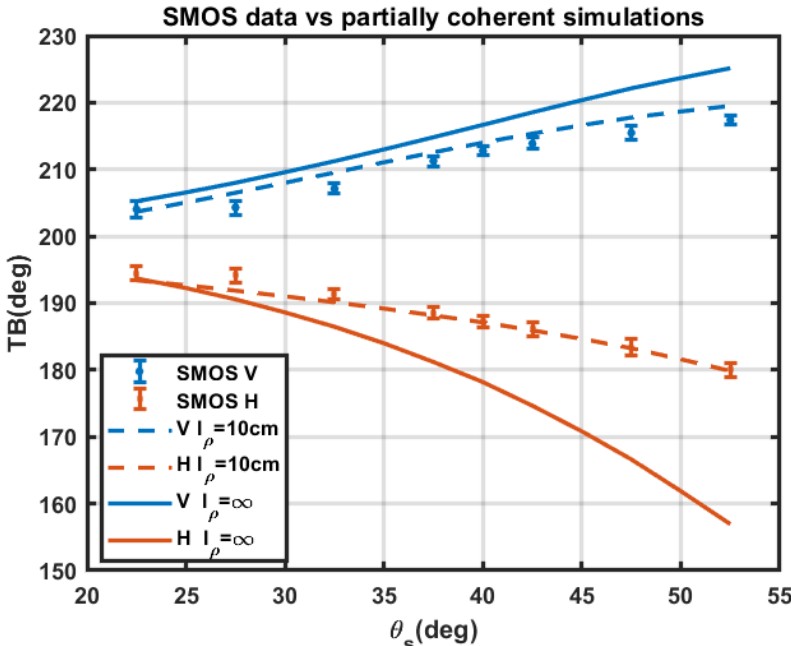

**Figure 14:** 10 year averaged SMOS data compared with partially coherent model forward simulations

Forward model predictions of 1.4 GHz brightness temperatures versus angle are shown in Figure 14 using $l_\rho = 10cm$ and

$l_\rho = \infty$, along with SMOS measurements averaged over a 10-year time period from 2011 to 2021. The SMOS error bars further indicate the expected accuracies of the SMOS data shown.

In simulating the results, the density parameters at z=0 are given by Table 8. The surface density fluctuation is selected between the ground measurements data at Dome C [Brogioni et al 2015] and the values obtained in [Leduc-Leballeur etal 2015]; note that information on the density correlation length is not provided in these works. The mean profile density follows

[Brogioni et al 2015] and a Robin-model temperature model is used. We assume that $std(\rho)$ and $l_z$ have dependence of $\exp(z/30)$ and $\exp(z/40)$ respectively to model the process of densification.

The results show that the model predictions with $l_\rho = 10cm$ provide good agreement with SMOS data in the range 22.5- 52.5 degrees incidence angle, and with RMS difference over angle of 1.4K in V and 0.8K in H. The one dimensional $l_\rho = \infty$ results in contrast show up to 17K differences in the H-pol simulations. These results show that including the effects of finite

 

horizontal correlation length allows the coupling between angle and polarization effects necessary to reproduce SMOS observations.

**Conclusions**

The results of the paper suggest a combined active and passive method for sensing long scale fluctuations in the firn density. These fluctuations contain information on accumulation and densification within the firn. The Community Firn Model was
used to generate profiles for comparison, and was shown to produce simulated profiles having reasonable agreement with in situ measurements provided that appropriate high resolution forcing data was available. Snow radar echo measurements were shown to provide information on refrozen layers within the firn, that could then be accounted for in analyzing 0.5-2 GHz brightness temperature datasets. The analytical partially coherent model reported was found to provide reasonable agreement with measured 0.5-2 GHz brightness temperatures by including the effects of refrozen layers and long scale density fluctuations.
Comparisons with SMOS measurements at Dome-C in particular demonstrate the coupling between H and V polarizations that is captured by the continuous random medium description used in the model. This work shows that the co-located active and passive microwave data can be used to infer the polar firn properties that can further be compared with predictions of CFM.

**Appendix A First order iterative approach for firn emission**

In this appendix, we give the details of the first order solution of radiative transfer equations with a varying mean and
fluctuating profile. The density profile is defined by:

$$\rho(z) = \rho_m(z) + \rho_f(z)$$

Where $\rho_m(z)$ is the mean profile which increases as the depth increases, $\rho_f(z)$ is the fluctuating profile with the standard deviation $std(\rho_f)(z)$ and vertical correlation length $l_z(z)$ decreases as z decreases. The radiative transfer equations for the density fluctuating region is given as:

$$cos\theta\frac{d}{dz}\vec{I}_u(\theta,z) = -\kappa_a(z)\vec{I}_u(\theta,z) + \kappa_a(z)CT_0 - \kappa_s(\theta,z)\vec{I}_u(\theta,z) + \int_0^{\frac{\pi}{2}}sin\theta'P_{uu}(\theta,\theta',z)\,\vec{I}_u(\theta',z)$$

$$+ \int_0^{\frac{\pi}{2}}sin\theta'P_{du}(\theta,\theta',z)\,\vec{I}_d(\theta',z)$$

And

$$-cos\theta\frac{d}{dz}\vec{I}_d(\theta,z) = -\kappa_a(z)\vec{I}_d(\theta,z) + \kappa_a(z)CT_0 - \kappa_s(\theta,z)\vec{I}_d(\theta,z) + \int_0^{\frac{\pi}{2}}sin\theta'P_{ud}(\theta,\theta',z)\,\vec{I}_u(\theta',z)$$

$$+ \int_0^{\frac{\pi}{2}}sin\theta'P_{dd}(\theta,\theta',z)\,\vec{I}_d(\theta',z)$$

The boundary conditions are given as the following:



$$\vec{I}_d(\theta, z = 0) = \vec{r}_{10}(\theta)\vec{I}_u(\theta, z = 0)$$

And

$$\vec{I}_u(\theta, z = -d) = CT_2$$

The intensity vectors contains the first and second components of the stokes vector. In the region we consider, which is about tens of meters below the surface, the physical temperature of the firn is almost a constant number, $T_0$.

The expressions for the phase functions can be found in [Tsang 2001].To find the solution, we multiply $\exp(-\int_{z'}^{0} \kappa_a(z'')sec\theta dz'')$ to the equation of upward going intensity and integrate the equation from $z' = -d$ to $z' = z$,after some math manipulations, we have the expressions for upward as:

$$\vec{I}_u(\theta, z) = CT_2 \exp\left(-\int_{-d}^{0} \kappa_a(z'')sec\theta dz''\right) + sec\theta \int_{-d}^{z} dz' \kappa_a(z')CT_0 \exp(-\int_{z'}^{z} \kappa_a(z'')sec\theta dz'')$$

$$- sec\theta \int_{-d}^{z} dz' \kappa_s(\theta, z')\vec{I}_u(\theta, z') \exp\left(-\int_{z'}^{z} \kappa_a(z'')sec\theta dz''\right)$$

$$+ \int_{-d}^{z} dz' \exp(-\int_{z'}^{z} \kappa_a(z'')sec\theta dz'' \left[\int_{0}^{\frac{\pi}{2}} P_{uu}(\theta, \theta', z')\vec{I}_u(\theta', z') + \int_{0}^{\frac{\pi}{2}} P_{du}(\theta, \theta', z')\vec{I}_d(\theta', z')\right]$$

for the downward intensity, we multiply the downward equation with $\exp(-\int_{-d}^{z'} \kappa_a(z'')sec\theta dz'')$ and integrate from $z' = z$ to $z' = 0$. The downward intensity is then obtained as:

$$\vec{I}_d(\theta, z) = \vec{r}_{10}(\theta)CT_2 \exp\left(-\int_{-d}^{0} \kappa_a(z'')sec\theta dz''\right) \exp\left(-\int_{z}^{0} \kappa_a(z'')sec\theta dz''\right)$$

$$+ \vec{r}_{10}(\theta) \exp\left(-\int_{z}^{0} \kappa_a(z'')sec\theta dz''\right) sec\theta \int_{-d}^{0} dz' \kappa_a(z')CT_0 \exp(-\int_{z'}^{0} \kappa_a(z'')sec\theta dz'')$$

$$+ sec\theta \int_{z}^{0} \kappa_a(z')CT_0 \exp(-\int_{z}^{z'} \kappa_a(z'')sec\theta dz'')dz'$$

$$- \vec{r}_{10}(\theta) \exp\left(-\int_{z}^{0} \kappa_a(z'')sec\theta dz''\right) sec\theta \int_{-d}^{0} dz' \kappa_s(\theta, z')\vec{I}_u(\theta, z') \exp\left(-\int_{z'}^{0} \kappa_a(z'')sec\theta dz''\right)$$

$$+ \vec{r}_{10}(\theta) \exp\left(-\int_{z}^{0} \kappa_a(z'')sec\theta dz''\right) sec\theta \int_{-d}^{0} dz' \exp(-\int_{-d}^{0} dz' \kappa_a(z')sec\theta dz') \left[\int_{0}^{\frac{\pi}{2}} sin\theta' P_{uu}(\theta, \theta', z')\vec{I}_u(\theta', z')\right.$$

$$\left. + \int_{0}^{\frac{\pi}{2}} sin\theta' P_{du}(\theta, \theta', z')\vec{I}_d(\theta', z')\right]$$

$$- sec\theta \int_{z}^{0} \kappa_s(\theta, z')\vec{I}_d(\theta, z') \exp\left(-\int_{z}^{z'} \kappa_a(z'')sec\theta dz''\right)dz'$$

$$+ sec\theta \int_{z}^{0} \exp(-\int_{z}^{z'} \kappa_a(z'')sec\theta dz'') \left[\int_{0}^{\frac{\pi}{2}} sin\theta' P_{ud}(\theta, \theta', z')\vec{I}_u(\theta', z') + \int_{0}^{\frac{\pi}{2}} sin\theta' P_{ud}(\theta, \theta', z')\vec{I}_d(\theta', z')\right]$$





The zeroth order solution for the upward and downward intensities are given as:

$$\vec{I}_u^{(0)}(\theta,z) = CT_2 \exp\left(-\int_{-d}^{0}\kappa_a(z'')sec\theta dz''\right) + sec\theta\int_{-d}^{z}dz'\kappa_a(z')CT_0\exp(-\int_{z'}^{z}\kappa_a(z'')sec\theta dz'')$$

And

$$\vec{I}_d^{(0)}(\theta,z) = \overleftrightarrow{r}_{10}(\theta)CT_2 \exp\left(-\int_{-d}^{0}\kappa_a(z'')sec\theta dz''\right)\exp\left(-\int_{z}^{0}\kappa_a(z'')sec\theta dz''\right)$$

$$+ \overleftrightarrow{r}_{10}(\theta)\exp\left(-\int_{z}^{0}\kappa_a(z'')sec\theta dz''\right)sec\theta\int_{-d}^{0}dz'\kappa_a(z')CT_0\exp(-\int_{z'}^{0}\kappa_a(z'')sec\theta dz'')$$

$$+ sec\theta\int_{z}^{0}\kappa_a(z')CT_0\exp(-\int_{z}^{z'}\kappa_a(z'')sec\theta dz'')dz'$$

The first order solution of the upward intensity is given as

$$\vec{I}_u^{(1)}(\theta,z) = -sec\theta\int_{-d}^{z}dz'\kappa_s(\theta,z')\vec{I}_u^{(0)}(\theta,z')\exp\left(-\int_{z'}^{z}\kappa_a(z'')sec\theta dz''\right)$$

$$+ sec\theta\int_{-d}^{0}dz'\exp(-\int_{z'}^{0}\kappa_a(z'')sec\theta dz'')\left[\int_{0}^{\frac{\pi}{2}}sin\theta' P_{uu}(\theta,\theta',z')\vec{I}_u(\theta',z')\right.$$

$$\left.+ \int_{0}^{\frac{\pi}{2}}sin\theta' P_{du}(\theta,\theta',z')\vec{I}_d(\theta',z')\right]$$

The specific intensity at $z = 0$ is then given as:

$$\vec{I}_u(\theta,z=0) = \vec{I}_u^{(0)}(\theta,z=0) + \vec{I}_u^{(1)}(\theta,z=0)$$

**Appendix B Robin model parameters for the 4 locations**

In this Appendix, present the robin model input parameters for the 4 locations

|  | Total ice thickness | Surface Temp | M | G |
|---|---|---|---|---|
| Pt1 | 2656 | 242.5 | 0.38 | 0.0886 |
| Pt2 | 3155 | 241.9 | 0.21 | 0.06 |
| Pt3 | 2951 | 241.5 | 0.235 | 0.095 |
| Pt4 | 3045 | 241.3 | 0.295 | 0.095 |

**Data availability**

The Morris density data can be provided upon request. UWBRAD data can be accessed from UWBRAD website.



**Author Contributions**

Haokui Xu did the major work. Brooke Medley provided all the simulations of density profiles and the map showing data positions over Greenland. Leung Tsang advised on the modelling. Joel.T.Johnson and Kenneth .C.Jezek provided UWBRAD data and valuable comments on the paper. Macro Brogioni, and Lars Kaleschke provide the SMOS data over Dome C and suggested the usage of data.

**Competing interests**

The authors declare that they have no conflict of interest.

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
