# Peer review of "Polar Firn Properties in Greenland and Antarctica and Related Effects on Microwave Brightness Temperatures"

_EGUsphere, 2022_

## Author Comment (AC1)

**Response to Reviewer comments 1**

*Comments from reviewer*

Responses

*This manuscript presents an exciting attempt to integrate active radar and passive microwave measurements to characterize firn properties (namely density) in both Greenland and Antarctica. The authors use airborne, high-bandwidth radar sounding data to constrain the presence of refrozen melt layers in firn and then use that information as an input to forward modeling of passive microwave brightness temperatures. The authors pull in a lot of different datasets (both modeled and observed) to provide a comprehensive analysis of near-surface firn properties.*

*Overall, while the scientific scope is extremely relevant and worthy of publication within The Cryosphere, I believe how it is presented and described within then manuscript would benefit from a thorough revision to 1) clarify/streamline the structure and 2) address how this research more clearly fits within the broader field.*

*Regarding the former, the authors make use of many different datasets and models (e.g., global climate model outputs, CFM, active radar, in situ measurements, UWBRAD, SMOS, forward model of brightness temperatures) but it isn't always clear how these pieces fit together to meet the main objective of the paper. To me, the manuscript is missing an overarching structure that the authors can use to guide the reader; the lack of which makes the manuscript difficult to follow.*

*Regarding the latter, the manuscript currently reads as a long Introduction followed by a flood of Results and then one paragraph of Conclusions. The authors do not make space for discussing the relevance of their results with respect to other work or how they see their work contributing to the field into the future. Currently, the authors leave all this to the reader to intuit for themselves, which can limit the impact the manuscript will have (i.e., be explicit and tell the reader why your work is important to them).*

*Below I have included suggestions for how the authors might consider revising their manuscript in light of these points as well as other comments and questions that I think would add to the manuscript.*

Thank you for the general comments. We will revise our paper according to the comments.

**Specific Comments**

*1) To address these issues raised in my General Comments, I'd recommend the authors consider adopting a more conventional manuscript structure (Introduction → Methods → Results → Discussion → Conclusions). As currently structured, the Introduction attempts to contextualize the research (which I think it does fairly well up until line 66) but then diverges into presenting*

*all the datasets and methodology to be followed through the remainder of the paper. I would recommend considering the inclusion of a stand-alone Methods section where the entire procedure the authors envision can be described (perhaps including a flowchart?) along with all the models (CFM, forward model) and datasets (in situ, Snow Radar, UWBRAD). I believe this would provide the necessary structure needed to organize the current Results section and a point of reference for a newly added "Discussion" section.*

Thank you for the comments on the structure of the paper. We will shorten the introduction parts and put most of the discussion of the procedure into the method part, such that the flow of the paper would be clearer.

We will adjust the paper to follow the structure of : Introduction → Methods → Results → Discussion → Conclusions.

We would add a "Discussion" section to discuss the relevance of the research and discuss the results presented in the paper. The major points will be discussed in the study relevance is listed below:

1. The density fluctuations show strong effects in Brightness temperature.
   a. Strong reduction on the UWBRAD TB
   b. Angular and polarization dependence in SMOS TB.
2. This work shows that passive microwave can be used as a tool to infer the density fluctuations remotely. There is no way to measure the density fluctuations except for in-situ measurement previously.
3. Understanding the density fluctuations is important in characterizing the mass balance of polar firn.
4. The radiative transfer model in this work can help interpret the TB data over Aquifer region.
5. Help reduce uncertainty in the ice sheet temperature profile retrieval.

*2) By their own admission (line 63), the authors are not the first to recognize that passive microwave measurements contain information on firn density fluctuations (note that Tan et al. (2019) does not appear in the reference list). However, there are noticeable absences from what the authors present as the existing research in this direction. How do the passive microwave Greenland density results of Houtz et al. (2019, 2021) and Mousavi et al. (2021) fit together with what is presented here?*

*Houtz et al. (2019) "Snow wetness and density retrieved from L-band satellite radiometer observations over a site in the West Greenland ablation zone" Remote Sensing of Environment 235 https://doi.org/10.1016/j.rse.2019.111361*

*Houtz et al. (2021) "Quantifying Surface Melt and Liquid Water on the Greenland Ice Sheet using L-band Radiometry" Remote Sensing of Environment 256 https://doi.org/10.1016/j.rse.2021.112341*

*Mousavi et lal. (2021) "Evaluation of Surface Melt on the Greenland Ice Sheet Using SMAP L-Band Microwave Radiometry" IEEE JSTARS 14 https://doi.org/10.1109/JSTARS.2021.3124229*

Thank you for referring us to these three research works. However, as the second reviewer pointed out, these works majorly focus their study on the lower elevation area of Greenland, which is, as indicated in the papers, close to the Coast of Greenland Island. The refrozen layers in the high elevation region is due to the great melt events.

*3) As a whole, the manuscript reads as under-cited. The authors often make statements that appear to require or refer to other sources without any indication of what those sources are. I have compiled the following list of example locations.*

*Line 35: "For example…"*

*Line 43: "Because the material…"*

*Line 45: "For example…"*

*Line 68: "In our previous…"*

*Line 132: "The second and third…" (specifically in reference to the NEGIS density profile)*

*Line 168: "UWBRAD measures…"*

*Line 171: "The University of…"*

*Line 204: "These peaks are…"*

*Line 242: "The exponential form…"*

*Line 249: "Although the top…"*

Thank you, we will add the references to the positions listed above.

Line 35:"For example, " add reference to  [Smith etal 2020].

Line 43:" Because the material"  add reference to [Smith etal 2020, Medely etal 2022]

Line 45:"For example…." Add reference to [Stevens 2020 etal].

Line 68:"In our previous works, we have used UWBRAD to sense the subsurface temperature profile….[Yardim etal 2022], "

C. Yardim *et al*., "Greenland Ice Sheet Subsurface Temperature Estimation Using Ultrawideband Microwave Radiometry," in *IEEE Transactions on Geoscience and Remote Sensing*, vol. 60, pp. 1-12, 2022, Art no. 4300312, doi: 10.1109/TGRS.2020.3043954.

Line 132 NEGIS:[Vallelonga etal 2014]

https://www1.ncdc.noaa.gov/pub/data/paleo/icecore/greenland/negis2012dens.txt

Vallelonga etal Initial results from geophysical surveys and shallow coring of the Northeast Greenland Ice Stream (NEGIS),The Cryosphere, 2014

Line 168 UWBRAD measures……[Andrews etal 2017]
M. Andrews *et al*., "The Ultra-Wideband Software Defined Microwave Radiometer (UWBRAD) for Ice sheet subsurface temperature sensing: Calibration and campaign results," *2017 IEEE International Geoscience and Remote Sensing Symposium (IGARSS)*, 2017, pp. 237-240, doi: 10.1109/IGARSS.2017.8126938.

Line 171 "The University of Kansas ….[Rodriguez etal 2010]"

Rodriguez-Morales, F., P. Gogineni, C. Leuschen, C. T. Allen, C. Lewis, A. Patel, L. Shi, W. Blake, B. Panzer, K. Byers, R. Crowe, L. Smith, and C. Gifford, Development of a Multi-Frequency Airborne Radar Instrumentation Package for Ice Sheet Mapping and Imaging, Proc. 2010 IEEE Int. Microwave Symp., Anaheim, CA, May 2010, pp. 157 – 160

Line 204 "These peaks are due to a melt evet that occurred in 2012 that affected much of Greenland." We modify this statement as "These peaks have amplitude comparable to the snow-air interface, which is likely to be caused due to the refrozen layers created in the melt events starting in the year of 2012."

Line 242"The exponential form …. [Tsang etal 2001]"

Line 249 This statement is due to the radiative transfer model. Consider the top 20m as a layer, the Brightness Temperature contributed from this part can be considered as:$T[1 - \exp(-\kappa_a d sec\theta)]$. The $\kappa_a$ is the absorption of firn, which is considered as very small. For a density of 0.3g/cm^3, using Matzler and Tiuri, the effecitive permittivity is 1.53 + 2e-5i, for

2GHz, the wavenumber is 42, which gives $\kappa_a = 7e - 4$. For d=20m, the exponential term gives 0.986. With a change of physical temperature of 30K, the resulting change in TB is 0.4K.

*4) In Section 2, the authors demonstrate the ability of the CFM to produce roughly equivalent density profiles to what has been measured in situ. I am left to wonder however, why these three specific in situ examples where chosen (i.e., Summit, NEEM and NEGIS)? The authors have already referred to the SUMup dataset (i.e., Montgomery et al., 2018) which contains many more in situ density profiles, some with better depth sampling than the NEEM and NEGIS examples and more contemporaneous with the UWBRAD and Snow Radar measurements. Is there a specific reason these three sites are preferred compared to others?*

Thanks for the question. Choosing the Summit data has a particular reason. This is because the high resolution provided by the neutron probe measurements (as high as 1cm vertical resolution) and its depth to 13 meters below the surface. This profile provides us the ability to evaluate the density fluctuation properties and enable us to compare the results from simulated profiles from Community Firn model.  Most of the density measurements provided in SUMup dataset are from snow pits. The sampling is usually on the scale of 10cm~1m, which cannot be used to fully characterize the variation. These variations are on the scale of the wavelength for UWBRAD. Variations on these scales have significant effects on the emissivity of the firn. Thus we need to use a fine sampled profile to evaluate these fluctuations on the microwave wavelength scale.

Besides, the conventional density measurements will inevitably sample the firn for a certain volume. This will cause an averaging effect to the measured density. CFM simulates the firn process for dry condition, where melt event is rare. Summit is near the center of Greenland, and thus is the most unlikely place to be affected by melt event.

NEEM and NEGIS provide a deep measurement of the density profile. We use these 2 data sets to compare with the mean profile of CFM results. Most of the snow pit measurements does not go beyond 10m depth.

*5) Table 1 presents a comparison of the Site 1 in situ density profile and the modeled CFM result. Variations in CFM density are constant with depth while the in situ ones decrease. At the same time, the vertical correlation lengths in the CFM results seem to be consistently smaller than those based on the in situ data. How are the metrics by which the "reasonable agreement" (line 166) between the CFM and in situ density profiles (i.e., standard deviations <0.03 (line 151) and correlation lengths <20 cm (line 152)) chosen? They seem rather arbitrary.*

*Furthermore, the caption states that one meter of data are used to estimate these properties but why is the depth interval at which these properties are reported two meters? Why not also*

*present the vertical density standard deviation and correlation length between one and two meters?*

The comparison shown here is to show that the community firn model is generating physical results that is comparable to the measurements and can be used as a reference to the input parameters in radiative transfer model. The community firn model is usually used to evaluate the mean density the firn. To our knowledge, CFM simulation results have not been used to study the density fluctuations. We want to show that CFM is giving physical results that is comparable to the measured density. 1.The simulated densities are not fluctuating with a very large amplitude, $(std(\rho) > mean(\rho))$ since observed density profile shows a rms density of fluctuation smaller than the mean profile. At this point the simulated profile and measured densities are in agreement.

2.The simulated profile is not changing too slow compared to the measurements. the mean correlation length of simulated profile is 2/3 of the measured profile. If the profile is changing too slow ($l_z^{CFM} > 2l_z^{measured}$), this means that the simulated CFM profile is not able to characterize the density changes.

We add a sentence in the manuscript to clarify from part one to explain the reasonable agreement

"The results from CFM usually used to evaluate the mean firn density. The comparison here is to show that 1. the CFM is not generating very large fluctuations $(std(\rho) > mean(\rho))$ since observed density profile shows a rms density of fluctuation smaller than the mean profile.2. The simulated profile is not changing too slow compared to the measurements. If the profile is changing too slow ($l_z^{CFM} > 2l_z^{measured}$), this means that the simulated CFM profile is not able to characterize the density changes. This shows that CFM results are reasonable compared to measured data. "

The data sets for the NEEM and NEGIS are sampled on the scale of 1 meter for the top 30 meters, please refer to the following links for NEEM and NEGIS:

NEEM:https://arcticdata.io/catalog/view/doi:10.18739/A2Q88G

NEGIS: https://www1.ncdc.noaa.gov/pub/data/paleo/icecore/greenland/negis2012dens.txt

The fluctuations from these 2 data sets are under sampled,

*6) The authors use "sites", "locations", and "points" when referring to individual measurements interchangeably. I would recommend using one term consistently through the entire manuscript. Furthermore, the specific sites (i.e., what is referred to as "Site 1") appear to change between sections. For example, in Section 2, "Site 1" refers to the location of the Summit in situ density measurement (line 136) but in Section 3, "Site 1" refers to the northernmost intersection of the UWBRAD and Snow Radar flight lines (Table 3, Figure 3). This is very confusing for the reader. I would recommend against Section-specific naming conventions.*

Thank you for pointing those out. We will use the term "points" instead of all the other terms.

*7) The authors have many figures that I believe could be condensed. For example, could Figures 1 and 2 be combined into a single figure with three sub-panels overlaying the measured and modelled density profiles? Could Figure 5 and Figure 6 be combined since they present different versions of the same information? Could Figures 9, 10, and 11 be combined into a single figure since they all share the same x-axis?*

Thanks for the suggestion, we will condense figure 1 and 2, figures 9 through 11. We will leave figure 5 and 6 as it is representing different physical quantities. We will improve the alignment of x axis for the simulated density profiles and the radar echoes.

*8) I recommend the authors clarify the relationship between their representation of the vertical density profiles used in Section 2 (line 126) and Section 4 (line 234). Is it necessary to include both a one-dimensional representation (i.e., Section 2) and the three-dimensional representation (i.e., Section 4)?*

The profiles in the 2 sections are different. Profiles shown in section 2 are field measurement data, which only provides a vertical representation of the density, the horizontal variation of the density could not be provided by these vertical measurements. In section 4, the 3-D representation of the density profile is implemented as our radiative transfer model. The variation in the horizontal scale is included. We will add the following sentence to clarify in section 4.

"Previous studies have treated the firn profile varying along the depth just as what is shown in the measured density profile. However, the angular dependence of TB in SMOS measurement cannot be explained by the 1D layered medium modelling [Tan etal 2015]. Horizontal variations should be considered."

*9) In Section 4, the authors introduce a horizontal correlation length for their density profile. What is the physical justification for including horizontal density variations? Furthermore, the emission model appears to be one-dimensional, so what does the additional horizontal density fluctuation contribute to the analysis?*

The emission model is 3-dimensional. We accounted for the azimuthal angle by integrating over $\phi_s$ since we have assumed the horizontal variation is symmetric. This is why in the radiative transfer equation, only integral over $\theta$ is shown.

The horizontal correlation length is representing the horizontal variations of the density profile. Microwave is sensitive to the horizontal variations in the centimeter scale. Horizontal variations will cause diffractions, coupling between vertical and horizontal pol. It will cause both angular and polarization dependence different from 1D layered media. This is critical for SMOS and SMAP, since SMOS have multi angle observation, and SMAP is 40 off nadir. Observations from SMOS show significant different in V and H for 1D layered medium model prediction.

*10) A key takeaway of this work seems to be the need for the co-acquisition of active radar and passive microwave data to asses any influence from refrozen layers. How do the authors perceive the broader applicability of their methodology moving forward knowing the current spatial and temporal coverage disparity between where active radar measurements (i.e., airborne at specific points in time) exist compared to passive microwave (i.e., satellites in continual operation)?*

A possible way is to study the time series space-borne microwave sensor data, passive or active, to obtain the historical information about melt events over the dry zone. The number of refrozen layers can thus be estimated and its effects can be evaluated. Then to infer the density, we can make use of the radiometer data sets operating at different frequencies.

*11) In Section 5, the authors attribute cross-frequency brightness temperature variations in the UWBRAD results between locations 1-3 and location 4 as the impact of more refrozen layers at location 4. Can they authors elaborate on why they believe it is solely the number of layers that affects the measured brightness temperature and not their position (i.e., the relative depth between layers as well as they're absolute position within the firn column)? Would it be more intuitive to expand on what is presented in Figure 11 to include the effects of multiple layers as well as their relative positioning.*

Considering the distances between the frozen layers meaning that we need to consider the coherent effects of electromagnetic waves. However, the variations in the density profile, both vertically and horizontally, will distort the phase relations of the reflected waves from each high density layers. Thus, we choose to consider the contribution of the reflectivity of each layer. We consider the coherent effects of the high density layer itself, since the thickness of the layer is small(~1cm) and definite.

*12) In Section 5, the authors introduce numerous exponential functions that are used to decrease different model parameters with depth (e.g., exp(z/33), exp(z/55), and exp(z/5) on line 315 and exp(z/30) and exp(z/40) on line 356). How were these specific functions chosen?*

We choose these factors to simulate a damping effect of the density fluctuation profiles. The selection of decay factor for density profile is to make sure that when then profile goes to the deeper part of the firn, the fluctuation will not make the density go beyond the ice density. The selection of decay factor for the correlation length choose to be slower than the decay of density to make sure that the density variation is negligible before the correlation length becomes very short (e.g. $l_z = 0.5cm$ with $std(\rho) = 0.01g/cm^3$).  The very fast decay one $\exp\left(\frac{z}{5}\right)$ is to model the decay of the profile affected by temporal variations. We choose it to have a fast decay since that the variation of this profile will be negligible due to densification.

*13) It is not clear why Antarctic SMOS data are included in the study. Why not use SMOS data from Greenland that overlap with the active radar and UWBRAD datasets? How does the Antarctic SMOS analysis contribute to findings of the paper?*

This part is trying to show the applicability of the radiative transfer model to the angular dependence of brightness temperature. The horizontal correlation length introduced in the model could explain the angular dependence of brightness temperature while the previous attempt using infinite layers can only explain v polarization of the brightness as in (Tan et al 2015).

*14) The authors repeatedly use "reasonable" as a qualitative, catch-all term for describing the agreement between two sets of data/observations without providing any justification. What specifically about these comparisons deem them to be indicative of a "reasonable agreement" from the authors perspective?*

*Line 164 (and line 166): The CFM results in Figure 2 do not reproduce the in situ variability at the top of the firn profile. What aspect of the CFM density profile is it that the authors are using to deem the agreement with the in situ profile reasonable? Is it simply the mean density?*

Figure 2 shows the mean density profile for these locations. We are trying to look at how the mean profile compare with the data from ice core.  This is why no fluctuations are seen. We will change the caption into "In-situ data and mean density profile from CFM for NEEM and NEGIS" Again, we want to show that the CFM results can be used as a reference.

*Line 239: What degree of vertical offset (i.e., +/-15 cm?) do the authors allow between the CFM density peaks and the active radar peaks for them to be illustrative of a reasonable agreement. How does this compare to the vertical resolution of these two datasets?*

We allow an offset of 60cm between the CFM density peak and the averaged radar echo as a reasonable agreement. Although the radar and the simulated density profiles have resolutions of 2cm and 1 cm respectively, it is not meaningful to compare the offset with resolutions. First of all, the distance of radar echograms as converted from the fast time of the data set. The subsurface permittivity is complicated which will affect the speed of EM waves travelling in the firn. A density needs to be assumed. We chose a density of 0.3g/cm^3 which is the density near surface. As it is shown either from the measurement or simulation, the density could be as high as to 0.4g/cm^3 and the surface density can be as small as to 0.2g/cm^3, which would correspond to a permittivity of 1.76 and 1.33 according to Matzler's permittivity model. For a given time of 17.6ns,  assuming 0.4g/cm^3 will give a 2 way distance of 2m while assuming 0.2g/cm^3 will give a 2-way distance of 2.6m.

*Line 311; When applying the iterative method to refine the model parameters, how do the authors define what a reasonable match is? Is there some error level, threshold, or similarity metric the authors use? If so, what is it?*

When using the passive microwave model, we control the overall rms error of brightness temperature within 3K. For a physical temperature of 240K, the rms error correspond to a relative error in the emissivity of 0.0125, which is usually of the value 0.8-0.9.

*15) Could the authors please clarify how the accumulation rates presented in Table 7 are calculated? Are they an output from the forward modelling of the brightness temperatures or are they simply calculated based on the depth to the assumed 2012 melt layer identified from the Snow Radar measurements?*

The accumulation rates are derived from the reanalysis-derived forcing data for the CFM. Thus, they do not use the snow radar data and are derived from an atmospheric model (see Medley et al., 2022).

***Technical Corrections***

*1) Please be consistent with referencing style. Sometimes references appear in square brackets while for others the authors use round brackets (i.e., [] vs ()). There are also some citations (e.g., line 237) with references that are missing publication years. Please follow the TC guidelines for citations.*

*2) Please ensure consistent symbols throughout the manuscript. For example, $\varepsilon$ is used in the equation on line 239 but $\in$ is used in lines 240 and 243.*

*3) Please ensure units are provided for every axis for every figure. There are some figures which have units for every axis (e.g., Figure 1), some with units for one axis only (e.g., Figure 2, Figure 4-7), and some without units completely (e.g., Figure 12).*

*4) Please modify Figure 4 as radar echograms are typically presented with distance along the x-axis and depth along the y-axis. Furthermore, I would recommend highlighting the exact portions of the echograms the individual profiles are pulled from. Finally, since the authors average echo profiles over one-kilometer sections, I would suggest denoting the x-axis of the modified echogram in groundtrack distance as opposed to latitude. I think this will be more intuitive for the reader as to how much averaging is done.*

*5) Please explicitly label subfigures (i.e., a, b, c, d, ...) instead of using positional cues (i.e., top-left, right-most, etc.).*

*6) Please follow the TC guidelines on numbering manuscript subsections in Section 5.*

*7) Please ensure that the CReSIS Snow **R**adar is capitalized when appropriate as it is the formal name for the system.*

*8) Please ensure the labelling and captions for each figure is correct. For example, the label on the right side of Figure 6 appears incorrect as well as the caption. I assume this should be point 4?*

*9) Please be consistent in how the amplitude of the active radar data are expressed. The echogram as part of Figure 4 presents the data in dB while all other plots (Figures 4-7) seem to present linear amplitudes. I would recommend plotting everything in dB as this is the more conventional representation for these data.*

*10) Please follow the TC guidelines with regards to labeling equations.*

*11) Please follow the TC guidelines with regards to unit notation (i.e., with negative exponents, non-italicized).*

*12) Line 247 makes reference to a "region 1" that seems to be associated with Figure 8; however there is no region 1 identified in Figure 8 so I am unsure to what the authors are referring.*

*13) Please use consistent notation in Table 5 as is used in the text (i.e., what is $corl_p$?). Also, please use a consistent font size within the table.*

*14) Please use a consistent Table style following the TC guidelines.*

*15) "frin" line 71*

*16) Instead of just the bandwidth, please provide the actual upper and lower frequency bounds for Snow Radar signals on line 84 as is done on line 172.*

*17) Units are missing for the depth in line 221.*

*18) There is an extra colon in line 233.*

*19) There is an extra colon in line 279.*

*20) Please provide units for the reflection amplitudes on line 280.*

*21) There is an extra colon in line 282.*

*22) There is a space missing on line 316 (i.e., "$asl_p$" compared to "as $l_p$").*

*23) Please fix the legend in for the Point 3 subplot in Figure 13 to match the rest of the subplots.*

Thank you for the comments on the technical problems in the paper. We will take care of these problems in the revised manuscript

---

## Author Comment (AC2)

**Response to Reviewer2**

*Comments from review*

Response

*General Comments*

*This is an interesting paper that combines the new Community Firn Model (CFM) with active (Snow Radar) and passive (UWBRAD and SMOS) microwave observations and an emission model to characterize firn stratigraphy at high-elevation sites in Greenland and Antarctica. The CFM is used to simulate density measurements. The SnowRadar is used to detect and then characterize high-density layers within the firn. Density profiles and high-density layers are used as inputs into the emission model. Model results are then compared to the microwave observations.*

*The topic and scope of the manuscript is relevant to the Cryosphere. However, as pointed out by Reviewer #1, the paper (1) lacks a coherent structure to guide the reader though the analysis, and (2) lacks a discussion about the results and the overall broader relevance of the study to the field. I would strongly suggest a major revision to improve the readability of the manuscript.*

*Specific Comments:*

*Reviewer #1 did an ~excellent job~ at pointing out most of the major issues in this paper in Specific Comments (1, 3-15).  I don't have too much more to add to this.*

*I would only disagree with Reviewer 1 on Specific Comments (2) – on detailed comparisons with Houtz et al. (2019, 2021) and Mousavi et al. (2021). Although there may be some similarities in the emission models, the focus of those studies is detecting meltwater in the lower-elevation percolation and ablation zone. This study focuses on high-elevation sites in what is typically the dry snow zone, which in some year's experiences extreme melt events. I don't think the comparison would be particularly relevant or useful.*

**1 -The introduction is too long, particularly compared with the length of the other sections and the overall manuscript.**

*I think much of the information in the introduction could more effectively be distributed into the main text. Specifically, following line 66 (In this paper…). For example, the details of the UWBRAD instrument, the CFM, and the Snow Radar. Following these descriptions, there is a relatively detailed description of the model.*

*A critical concept that is somewhat missed in the introduction, and a significant strength of the method, is the concept of refreezing high-density layers. Over the last decade, extreme melt events in the interior of Greenland have become more frequent, with melting detected at Summit in 2012, 2019, 2021(including rain!). These melting trends will likely continue, which will*

*routinely bury high-density layers in the firn, and ultimately alter the interior structure of the ice sheet, which has mass balance implications.*

*From the perspective of EM modeling, typical dry snow models with layered firn will need to be adapted to account for these high-density layers, which can range from simple ice layers (this paper), to layers formed via shallow or deep vertical percolation of meltwater, with larger, vertically distributed ice structures (e.g.,*

> *10. C. Jezek et al., "500–2000-MHz Brightness Temperature Spectra of the Northwestern Greenland Ice Sheet," in IEEE Transactions on Geoscience and Remote Sensing, vol. 56, no. 3, pp. 1485-1496, March 2018, doi: 10.1109/TGRS.2017.2764381.)*

*Line 56 states: The strongest echoes in a radar echogram, for example, show the position of abrupt permittivity changes that usually correspond to the position of refrozen melt layers (Jezek and others, 1994; Zabel and others, 1995). An alternate or additional reference with high-density layers that are closer in structure to what you might find in cross-over sites is*

*Culberg, R., Schroeder, D.M. & Chu, W. Extreme melt season ice layers reduce firn permeability across Greenland. Nat Commun **12**, 2336 (2021). https://doi.org/10.1038/s41467-021-22656-5.*

Thank you for the comments. As in the response to the first reviewer, we will shorten the introduction part of the paper and put the detailed information into the "Method" section.

Thank you for the suggestion on the refrozen layers. This is a really important part of our work. The previous microwave radiometry models have not included the effects of these refrozen layers for the high elevation area in Greenland.  We will include the information provided by the reviewer and emphasize the inclusion of refrozen layers.

*2 -The manuscript structure is difficult to follow.*

*I agree with Reviewer #1's suggestion for a more formal structure: Introduction → Methods → Results → Discussion → Conclusions. Some suggestions: I might start with a flowchart linking models with the data sets. I might next introduce the model – which nicely provides the emission concept (Fig. 8) and instantly clarifies to the reader the objective. I might then follow with the details of the input data. Then the model results. Then comparisons with UWBRAD data for Greenland only (#3 below). Then a strong discussion which is currently missing from the manuscript.*

Thank you for the comments on the flow of the paper. We will revise the paper according to the suggested formal structure as Introduction → Methods → Results → Discussion → Conclusions. As responded to the first reviewer and in comments 1, we will shorten the introduction and put the details in the "method" section.

*3 -The paper would be much stronger if it focused on just Greenland.*

*The Antarctica comparison seems out of place in the manuscript. The paper starts out with a model that includes high-density layers, and data from the CFM, the SNOW RADAR, and UWBRAD over Greenland. The paper then shifts gear to Dome C, a site where high-density layers do not form, and a model comparison with a different instrument (SMOS). Perhaps the general idea was to compare sites with different firn characteristics. If that were the case, it would be more straightforward from the perspective of the reader to include a UWBRAD comparison between sites (UWBRAD data was collected at Dome C) or a SMOS comparison between sites. But, I don't think that this comparison is needed for this manuscript, Greenland, with a strong discussion section, is sufficient.*

Thank you for the comments. We will put this result to the discussion section as a broader relevance to the field

The horizontal correlation provides a tool to interpret the V and H angular dependence brightness temperature. It can also help to interpret the SMAP V and H brightness temperature data over the region with perennial aquifer.

.

*4 - The manuscript lacks a discussion section that describes the study relevance.*

Thank you for pointing that out again. We will add a discussion section to the manuscript. The major points will be discussed in the study relevance is listed below:

1. The density fluctuations show strong effects in Brightness temperature.
    a. Strong reduction on the UWBRAD TB
    b. Angular and polarization dependence in SMOS TB.
2. This work shows that passive microwave can be used as a tool to infer the density fluctuations remotely. There is no way to measure the density fluctuations except for in-situ measurement previously.
3. Understanding the density fluctuations is important in characterizing the mass balance of polar firn.
4. The radiative transfer model in this work can help interpret the TB data over Aquifer region.
5. Help reduce uncertainty in the ice sheet temperature profile retrieval.

*Technical Corrections*

*Line 14 - locations in the Greenland Ice Sheet - > locations on the Greenland Ice Sheet*

*Line 14 - and at the Dome C location - > and at Dome C*

*Line 15 - Borehole in situ measurements - > Borehole measurements*

*Line 65 - Kirchhoff's Law[Tsang 2001] - > add space*

*Line 131 - T41(71.08N,37.92W) - > add space*

*Line 140 - Summit station, Greenland - > Summit Station, Greenland*

*Line 147 - from Neutron Probe of Morris and Wingham, 2011 - >*

    *from the Neutron Probe Morris and Wingham, (2011)*

*Line 153 - 9.4cm - > space*

*Line 160 - In-situ Measurements - > in situ measurements*

*Line 180 - X-Ray - > X-ray, also throughout text*

*Line 184 – Table 3: Latitude and Longitude for crossover points of 2017 UWBRAD and Snow Radar Measurements - > Table 3: Latitude and longitude for crossover points of 2017 UWBRAD and Snow Radar measurements*

*Figure 4, 5, 6 – Sizes of plots are different (Fig. 4) – please correct. Reverse x- and y- axes, so density is vertical, which is the typical orientation.*

*Line 276 - Tan et al 2020 – Tan et al., (2020)*

*Line 279 - Figure 9: - > remove colon*

*Line 282 - Figure 10: - > remove colon*

*Lines 290 - ð• Ÿ• . ð• Ÿ" ð• Ÿ' (0.35g/cm^3 in density) - > add space*

*There are many places with issues with spaces (or lack of spaces), punctuation (especially random colons), notation, un-needed capitalization (particularly in table and figure captions). Please give the manuscript a ~very careful review~ for these issues during the revision.*

Thank you for the careful review of our paper. We will attend to all these issues in the revision.

---

## Author Response (AR2)

The technical corrections are listed here and marked in red also in the Manuscript

Line 13:

In studying the mass balance of polar ice sheets, the fluctuation of the firn density near the surface is a major uncertainty.

Is changed into:

In studying the mass balance of polar ice sheets, fluctuations of firn density near the surface is a major uncertainty.

Line 57:
The strongest echoes in a radar echogram, for example, show the position of abrupt permittivity changes that usually correspond to the position of refrozen melt layers (Jezek and others, 1994; Zabel and others, 1995).

Into :

The strongest echoes in a radar echogram, for example, show the positions of abrupt permittivity changes that usually correspond to the positions of refrozen melt layers (Jezek and others, 1994; Zabel and others, 1995).

Line 64

Change " In our previous works [Yardim etal 20222], we have used UWBRAD to sense the subsurface temperature profile, in which case the reflections caused by firn density fluctuations are nuisance effects. "

Into:

"In our previous work, we have used Ultra-Wide Band software defined RADiometer (UWBRAD) to sense the subsurface temperature profile, in which case the reflections caused by firn density fluctuations are nuisance effects[Yardim etal 20222]."

Line 69:

[revised manuscript text omitted]

Table 6 and 7.

The first rows of these 2 tables are changed into "Location 1, Location 2, Location 3 Location 4 "

Line 325 to Line 330

Long scale density fluctuation parameters inferred from the CFM……….

The "sites" and "Pts" are changed into "Locations"